# PLAYING SNES IN THE RETRO LEARNING ENVIRONMENT

**Nadav Bhonker\*, Shai Rozenberg\* and Itay Hubara**
Department of Electrical Engineering
Technion, Israel Institute of Technology
(\*) indicates equal contribution
{nadavbh,shairoz}@tx.technion.ac.il
itayhubara@gmail.com

## ABSTRACT

Mastering a video game requires skill, tactics and strategy. While these attributes may be acquired naturally by human players, teaching them to a computer program is a far more challenging task. In recent years, extensive research was carried out in the field of reinforcement learning and numerous algorithms were introduced, aiming to learn how to perform human tasks such as playing video games. As a result, the Arcade Learning Environment (ALE) (Bellemare et al., 2013) has become a commonly used benchmark environment allowing algorithms to train on various Atari 2600 games. In many games the state-of-the-art algorithms outperform humans. In this paper we introduce a new learning environment, the Retro Learning Environment — RLE, that can run games on the Super Nintendo Entertainment System (SNES), Sega Genesis and several other gaming consoles. The environment is expandable, allowing for more video games and consoles to be easily added to the environment, while maintaining the same interface as ALE. Moreover, RLE is compatible with Python and Torch. SNES games pose a significant challenge to current algorithms due to their higher level of complexity and versatility.

## 1 INTRODUCTION

Controlling artificial agents using only raw high-dimensional input data such as image or sound is a difficult and important task in the field of Reinforcement Learning (RL). Recent breakthroughs in the field allow its utilization in real-world applications such as autonomous driving (Shalev-Shwartz et al., 2016), navigation (Bischoff et al., 2013) and more. Agent interaction with the real world is usually either expensive or not feasible, as the real world is far too complex for the agent to perceive. Therefore in practice the interaction is simulated by a virtual environment which receives feedback on a decision made by the algorithm. Traditionally, games were used as a RL environment, dating back to Chess (Campbell et al., 2002), Checkers (Schaeffer et al., 1992), backgammon (Tesauro, 1995) and the more recent Go (Silver et al., 2016). Modern games often present problems and tasks which are highly correlated with real-world problems. For example, an agent that masters a racing game, by observing a simulated driver's view screen as input, may be usefull for the development of an autonomous driver. For high-dimensional input, the leading benchmark is the Arcade Learning Environment (ALE) (Bellemare et al., 2013) which provides a common interface to dozens of Atari 2600 games, each presents a different challenge. ALE provides an extensive benchmarking platform, allowing a controlled experiment setup for algorithm evaluation and comparison. The main challenge posed by ALE is to successfully play as many Atari 2600 games as possible (i.e., achieving a score higher than an expert human player) without providing the algorithm any game-specific information (i.e., using the same input available to a human - the game screen and score). A key work to tackle this problem is the Deep Q-Networks algorithm (Mnih et al., 2015), which made a breakthrough in the field of Deep Reinforcement Learning by achieving human level performance on 29 out of 49 games. In this work we present a new environment — the Retro Learning Environment (RLE). RLE sets new challenges by providing a unified interface for Atari 2600 games as well as more advanced gaming consoles. As a start we focused on the Super Nintendo Entertainment

System (SNES). Out of the five SNES games we tested using state-of-the-art algorithms, only one was able to outperform an expert human player. As an additional feature, RLE supports research of multi-agent reinforcement learning (MARL) tasks (Buşoniu et al., 2010). We utilize this feature by training and evaluating the agents against each other, rather than against a pre-configured in-game AI. We conducted several experiments with this new feature and discovered that agents tend to learn how to overcome their current opponent rather than generalize the game being played. However, if an agent is trained against an ensemble of different opponents, its robustness increases. The main contributions of the paper are as follows:

- Introducing a novel RL environment with significant challenges and an easy agent evaluation technique (enabling agents to compete against each other) which could lead to new and more advanced RL algorithms.

- A new method to train an agent by enabling it to train against several opponents, making the final policy more robust.

- Encapsulating several different challenges to a single RL environment.

## 2 RELATED WORK

### 2.1 ARCADE LEARNING ENVIRONMENT

The Arcade Learning Environment is a software framework designed for the development of RL algorithms, by playing Atari 2600 games. The interface provided by ALE allows the algorithms to select an action and receive the Atari screen and a reward in every step. The action is the equivalent to a human's joystick button combination and the reward is the difference between the scores at time stamp $t$ and $t - 1$. The diversity of games for Atari provides a solid benchmark since different games have significantly different goals. Atari 2600 has over 500 games, currently over 70 of them are implemented in ALE and are commonly used for algorithm comparison.

### 2.2 INFINITE MARIO

Infinite Mario (Togelius et al., 2009) is a remake of the classic Super Mario game in which levels are randomly generated. On these levels the Mario AI Competition was held. During the competition, several algorithms were trained on Infinite Mario and their performances were measured in terms of the number of stages completed. As opposed to ALE, training is not based on the raw screen data but rather on an indication of Mario's (the player's) location and objects in its surrounding. This environment no longer poses a challenge for state of the art algorithms. Its main shortcoming lie in the fact that it provides only a single game to be learnt. Additionally, the environment provides hand-crafted features, extracted directly from the simulator, to the algorithm. This allowed the use of planning algorithms that highly outperform any learning based algorithm.

### 2.3 OPENAI GYM

The OpenAI gym (Brockman et al., 2016) is an open source platform with the purpose of creating an interface between RL environments and algorithms for evaluation and comparison purposes. OpenAI Gym is currently very popular due to the large number of environments supported by it. For example *ALE, Go, MouintainCar* and *VizDoom* (Zhu et al., 2016), an environment for the learning of the 3D first-person-shooter game "Doom". OpenAI Gym's recent appearance and wide usage indicates the growing interest and research done in the field of RL.

### 2.4 OPENAI UNIVERSE

Universe (Universe, 2016) is a platform within the OpenAI framework in which RL algorithms can train on over a thousand games. Universe includes very advanced games such as *GTA V, Portal* as well as other tasks (e.g. browser tasks). Unlike RLE, Universe doesn't run the games locally and requires a VNC interface to a server that runs the games. This leads to a lower frame rate and thus longer training times.

## 2.5 Malmo

Malmo (Johnson et al., 2016) is an artificial intelligence experimentation platform of the famous game *"Minecraft"*. Although Malmo consists of only a single game, it presents numerous challenges since the *"Minecraft"* game can be configured differently each time. The input to the RL algorithms include specific features indicating the "state" of the game and the current reward.

## 2.6 DeepMind Lab

DeepMind Lab (Dee) is a first-person 3D platform environment which allows training RL algorithms on several different challenges: static/random map navigation, collect fruit (a form of reward) and a laser-tag challenge where the objective is to tag the opponents controlled by the in-game AI. In LAB the agent observations are the game screen (with an additional depth channel) and the velocity of the character. LAB supports four games (one game - four different modes).

## 2.7 Deep Q-Learning

In our work, we used several variant of the Deep Q-Network algorithm (DQN) (Mnih et al., 2015), an RL algorithm whose goal is to find an optimal policy (i.e., given a current state, choose action that maximize the final score). The state of the game is simply the game screen, and the action is a combination of joystick buttons that the game responds to (i.e., moving ,jumping). DQN learns through trial and error while trying to estimate the "Q-function", which predicts the cumulative discounted reward at the end of the episode given the current state and action while following a policy $\pi$. The Q-function is represented using a convolution neural network that receives the screen as input and predicts the best possible action at it's output. The Q-function weights $\theta$ are updated according to:

$$\theta_{t+1}(s_t, a_t) = \theta_t + \alpha(R_{t+1} + \gamma \max_a(Q_t(s_{t+1}, a; \theta'_t)) - Q_t(s_t, a_t; \theta_t))\nabla_\theta Q_t(s_t, a_t; \theta_t), \quad (1)$$

where $s_t$, $s_{t+1}$ are the current and next states, $a_t$ is the action chosen, $\alpha$ is the step size, $\gamma$ is the discounting factor $R_{t+1}$ is the reward received by applying $a_t$ at $s_t$. $\theta'$ represents the previous weights of the network that are updated periodically. Other than DQN, we examined two leading algorithms on the RLE: Double Deep Q-Learning (D-DQN) (Van Hasselt et al., 2015), a DQN based algorithm with a modified network update rule. Dueling Double DQN (Wang et al., 2015), a modification of D-DQN's architecture in which the Q-function is modeled using a state (screen) dependent estimator and an action dependent estimator.

# 3 THE RETRO LEARNING ENVIRONMENT

## 3.1 Super Nintendo Entertainment System

The Super Nintendo Entertainment System (SNES) is a home video game console developed by Nintendo and released in 1990. A total of 783 games were released, among them, the iconic *Super Mario World*, *Donkey Kong Country* and *The Legend of Zelda*. Table (1) presents a comparison between Atari 2600, Sega Genesis and SNES game consoles, from which it is clear that SNES and Genesis games are far more complex.

## 3.2 Implementation

To allow easier integration with current platforms and algorithms, we based our environment on the ALE, with the aim of maintaining as much of its interface as possible. While the ALE is highly coupled with the Atari emulator, Stella[1], RLE takes a different approach and separates the learning environment from the emulator. This was achieved by incorporating an interface named LibRetro (libRetro site), that allows communication between front-end programs to game-console emulators. Currently, LibRetro supports over 15 game consoles, each containing hundreds of games, at an estimated total of over 7,000 games that can potentially be supported using this interface. Examples of supported game consoles include *Nintendo Entertainment System, Game Boy, N64, Sega Genesis,*

---

[1]http://stella.sourceforge.net/

*Saturn, Dreamcast and Sony PlayStation.* We chose to focus on the SNES game console implemented using the snes9x[2] as it's games present interesting, yet plausible to overcome challenges. Additionally, we utilized the Genesis-Plus-GX[3] emulator, which supports several Sega consoles: Genesis/Mega Drive, Master System, Game Gear and SG-1000.

## 3.3 SOURCE CODE

RLE is fully available as open source software for use under GNU's General Public License[4]. The environment is implemented in C++ with an interface to algorithms in C++, Python and Lua. Adding a new game to the environment is a relatively simple process.

## 3.4 RLE INTERFACE

RLE provides a unified interface to all games in its supported consoles, acting as an RL-wrapper to the LibRetro interface. Initialization of the environment is done by providing a game (ROM file) and a gaming-console (denoted by 'core'). Upon initialization, the first state is the initial frame of the game, skipping all menu selection screens. The cores are provided with the RLE and installed together with the environment. Actions have a bit-wise representation where each controller button is represented by a one-hot vector. Therefore a combination of several buttons is possible using the bit-wise OR operator. The number of valid buttons combinations is larger than 700, therefore only the meaningful combinations are provided. The environments observation is the game screen, provided as a 3D array of 32 bit per pixel with dimensions which vary depending on the game. The reward can be defined differently per game, usually we set it to be the score difference between two consecutive frames. By setting different configuration to the environment, it is possible to alter in-game properties such as difficulty (i.e easy, medium, hard), its characters, levels, etc.

Table 1: Atari 2600, SNES and Genesis comparison

|  | **Atari 2600** | **SNES** | **Genesis** |
| --- | --- | --- | --- |
| Number of Games | 565 | 783 | 928 |
| CPU speed | 1.19MHz | 3.58MHz | 7.6 MHz |
| ROM size | 2-4KB | 0.5-6MB | 16 MBytes |
| RAM size | 128 bytes | 128KB | 72KB |
| Color depth | 8 bit | 16 bit | 16 bit |
| Screen Size | 160x210 | 256x224 or 512x448 | 320x224 |
| Number of controller buttons | 5 | 12 | 11 |
| Possible buttons combinations | 18 | over 720 | over 100 |

## 3.5 ENVIRONMENT CHALLENGES

Integrating SNES and Genesis with RLE presents new challenges to the field of RL where visual information in the form of an image is the only state available to the agent. Obviously, SNES games are significantly more complex and unpredictable than Atari games. For example in sports games, such as NBA, while the player (agent) controls a single player, all the other nine players' behavior is determined by pre-programmed agents, each exhibiting random behavior. In addition, many SNES games exhibit delayed rewards in the course of their play (i.e., reward for an actions is given many time steps after it was performed). Similarly, in some of the SNES games, an agent can obtain a reward that is indirectly related to the imposed task. For example, in platform games, such as *Super Mario*, reward is received for collecting coins and defeating enemies, while the goal of the challenge is to reach the end of the level which requires to move to keep moving to the right. Moreover, upon completing a level, a score bonus is given according to the time required for its completion. Therefore collecting coins or defeating enemies is not necessarily preferable if it consumes too much time. Analysis of such games is presented in section 4.2. Moreover, unlike Atari that consists of

---

[2]http://www.snes9x.com/

[3]https://github.com/ekeeke/Genesis-Plus-GX

[4]https://github.com/nadavbh12/Retro-Learning-Environment

eight directions and one action button, SNES has eight-directions pad and six actions buttons. Since combinations of buttons are allowed, and required at times, the actual actions space may be larger than 700, compared to the maximum of 18 actions in Atari. Furthermore, the background in SNES is very rich, filled with details which may move locally or across the screen, effectively acting as non-stationary noise since it provided little to no information regarding the state itself. Finally, we note that SNES utilized the first 3D games. In the game *Wolfenstein*, the player must navigate a maze from a first-person perspective, while dodging and attacking enemies. The SNES offers plenty of other 3D games such as flight and racing games which exhibit similar challenges. These games are much more realistic, thus inferring from SNES games to "real world" tasks, as in the case of self driving cars, might be more beneficial. A visual comparison of two games, Atari and SNES, is presented in Figure (1).

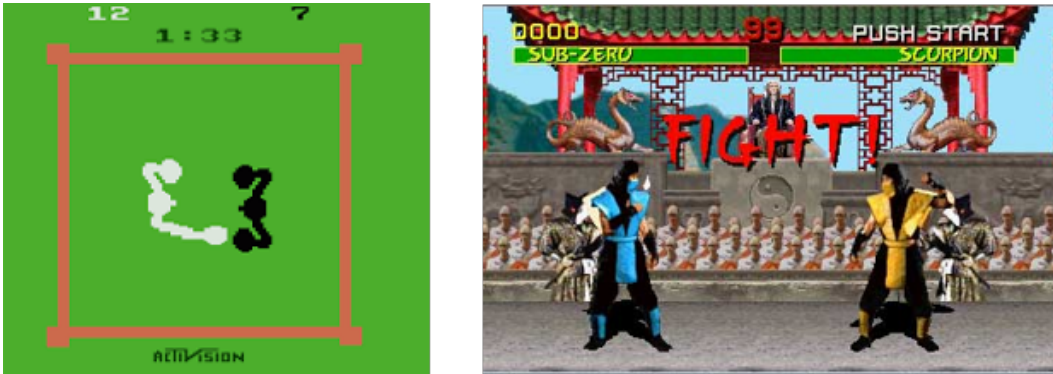

Figure 1: Atari 2600 and SNES game screen comparison: **Left:** "Boxing" an Atari 2600 fighting game , **Right:** "Mortal Kombat" a SNES fighting game. Note the exceptional difference in the amount of details between the two games. Therefore, distinguishing a relevant signal from noise is much more difficult.

Table 2: Comparison between RLE and the latest RL environments

| Characteristics | RLE | OpenAI Universe | Inifinte Mario | ALE | Project Malmo | DeepMind Lab |
|---|---|---|---|---|---|---|
| Number of Games | 8 out of 7000+ | 1000+ | 1 | 74 | 1 | 4 |
| In game adjustments[1] | Yes | NO | No | No | Yes | Yes |
| Frame rate | 530fps[2](SNES) | 60fps | 5675fps[2] | 120fps | <7000fps | <1000fps |
| Observation (Input) | screen, RAM | Screen | hand crafted features | screen, RAM | hand crafted features | screen + depth and velocity |

[1] Allowing changes in-the game configurations (e.g., changing difficulty, characters, etc.)

[2] Measured on an i7-5930k CPU

## 4 EXPERIMENTS

### 4.1 EVALUATION METHODOLOGY

The evaluation methodology that we used for benchmarking the different algorithms is the popular method proposed by (Mnih et al., 2015). Each examined algorithm is trained until either it reached convergence or 100 epochs (each epoch corresponds to 50,000 actions), thereafter it is evaluated by performing 30 episodes of every game. Each episode ends either by reaching a terminal state or after 5 minutes. The results are averaged per game and compared to the average result of a human player. For each game the human player was given two hours for training, and his performances were evaluated over 20 episodes. As the various algorithms don't use the game audio in the learning process, the audio was muted for both the agent and the human. From both, humans and agents

score, a random agent score (an agent performing actions randomly) was subtracted to assure that learning indeed occurred. It is important to note that DQN's $\epsilon$-greedy approach (select a random action with a small probability $\epsilon$) is present during testing thus assuring that the same sequence of actions isn't repeated. While the screen dimensions in SNES are larger than those of Atari, in our experiments we maintained the same pre-processing of DQN (i.e., downscaling the image to 84x84 pixels and converting to gray-scale). We argue that downscaling the image size doesn't affect a human's ability to play the game, therefore suitable for RL algorithms as well. To handle the large action space, we limited the algorithm's actions to the minimal button combinations which provide unique behavior. For example, on many games the R and L action buttons don't have any use therefore their use and combinations were omitted.

### 4.1.1 RESULTS

A thorough comparison of the four different agents' performances on SNES games can be seen in Figure (). The full results can be found in Table (3). Only in the game *Mortal Kombat* a trained agent was able to surpass a expert human player performance as opposed to Atari games where the same algorithms have surpassed a human player on the vast majority of the games.

One example is *Wolfenstein* game, a 3D first-person shooter game, requires solving 3D vision tasks, navigating in a maze and detecting object. As evident from figure (2), all agents produce poor results indicating a lack of the required properties. By using $\epsilon$-greedy approach the agents weren't able to explore enough states (or even other rooms in our case). The algorithm's final policy appeared as a random walk in a 3D space. Exploration based on visited states such as presented in Bellemare et al. (2016) might help addressing this issue. An interesting case is Gradius III, a side-scrolling, flight-shooter game. While the trained agent was able to master the technical aspects of the game, which includes shooting incoming enemies and dodging their projectiles, it's final score is still far from a human's. This is due to a hidden game mechanism in the form of "power-ups", which can be accumulated, and significantly increase the players abilities. The more power-ups collected without being use — the larger their final impact will be. While this game-mechanism is evident to a human, the agent acts myopically and uses the power-up straight away[5].

### 4.2 REWARD SHAPING

As part of the environment and algorithm evaluation process, we investigated two case studies. First is a game on which DQN had failed to achieve a better-than-random score, and second is a game on which the training duration was significantly longer than that of other games.

In the first case study, we used a 2D back-view racing game "F-Zero". In this game, one is required to complete four laps of the track while avoiding other race cars. The reward, as defined by the score of the game, is only received upon completing a lap. This is an extreme case of a reward delay. A lap may last as long as 30 seconds, which span over 450 states (actions) before reward is received. Since DQN's exploration is a simple $\epsilon$-greedy approach, it was not able to produce a useful strategy. We approached this issue using reward shaping, essentially a modification of the reward to be a function of the reward and the observation, rather than the reward alone. Here, we define the reward to be the sum of the score and the agent's speed (a metric displayed on the screen of the game). Indeed when the reward was defined as such, the agents learned to finish the race in first place within a short training period.

The second case study is the famous game of Super Mario. In this game the agent, Mario, is required to reach the right-hand side of the screen, while avoiding enemies and collecting coins. We found this case interesting as it involves several challenges at once: dynamic background that can change drastically within a level, sparse and delayed rewards and multiple tasks (such as avoiding enemies and pits, advancing rightwards and collecting coins). To our surprise, DQN was able to reach the end of the level without any reward shaping, this was possible since the agent receives rewards for events (collecting coins, stomping on enemies etc.) that tend to appear to the right of the player, causing the agent to prefer moving right. However, the training time required for convergence was significantly longer than other games. We defined the reward as the sum of the in-game reward and a bonus granted according the the player's position, making moving right preferable. This reward

---

[5]A video demonstration can be found at https://youtu.be/nUl9XLMveEU

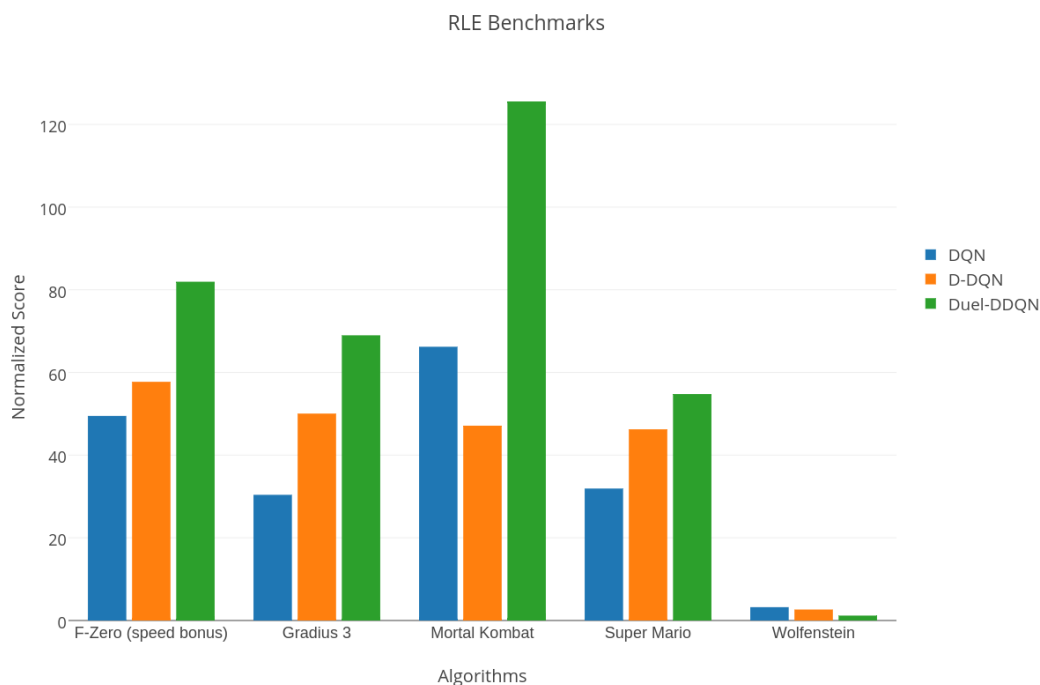

Figure 2: DQN, DDQN and Duel-DDQN performance. Results were normalized by subtracting the a random agent's score and dividing by the human player score. Thus 100 represents a human player and zero a random agent.

proved useful, as training time required for convergence decreased significantly. The two games above can be seen in Figure (3).

Figure (4) illustrates the agent's average value function . Though both were able complete the stage trained upon, the convergence rate with reward shaping is significantly quicker due to the immediate realization of the agent to move rightwards.

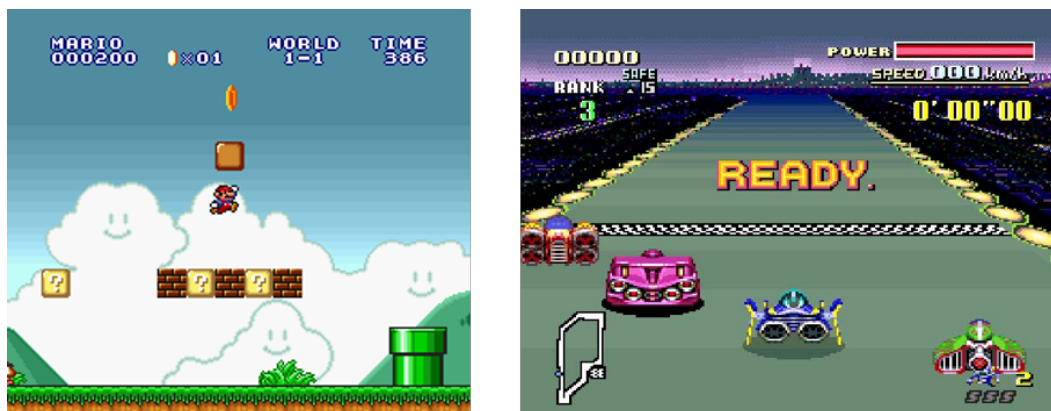

Figure 3: **Left:** The game *Super Mario* with added bonus for moving right, enabling the agent to master them game after less training time. **Right:** The game *F-Zero*. By granting a reward for speed the agent was able to master this game, as oppose to using solely the in-game reward.

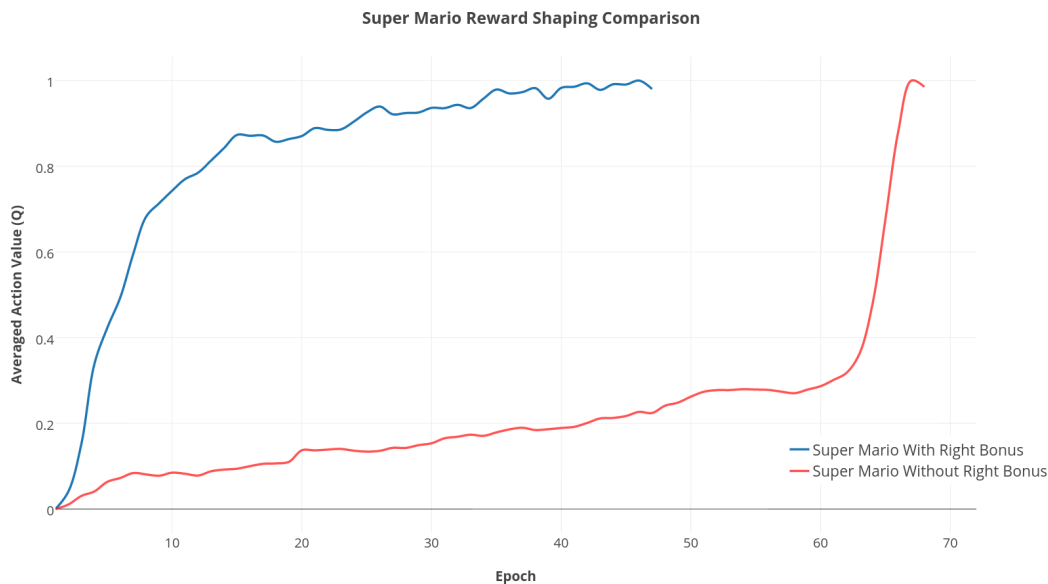

Figure 4: Averaged action-value (Q) for Super Mario trained with reward bonus for moving right (blue) and without (red).

### 4.3 MULTI-AGENT REINFORCEMENT LEARNING

In this section we describe our experiments with RLE's multi-agent capabilities. We consider the case where the number of agents, $n = 2$ and the goals of the agents are opposite, as in $r_1 = -r_2$. This scheme is known as fully competitive (Buşoniu et al., 2010). We used the simple single-agent RL approach (as described by Buşoniu et al. (2010) section 5.4.1) which is to apply to single agent approach to the multi-agent case. This approach was proved useful in Crites and Barto (1996) and Matarić (1997). More elaborate schemes are possible such as the minimax-Q algorithm (Littman, 1994), (Littman, 2001). These may be explored in future works. We conducted three experiments on this setup: the first use was to train two different agents against the in-game AI, as done in previous sections, and evaluate their performance by letting them compete against each other. Here, rather than achieving the highest score, the goal was to win a tournament which consist of 50 rounds, as common in human-player competitions. The second experiment was to initially train two agents against the in-game AI, and resume the training while competing against each other. In this case, we evaluated the agent by playing again against the in-game AI, separately. Finally, in our last experiment we try to boost the agent capabilities by alternated it's opponents, switching between the in-game AI and other trained agents.

### 4.3.1 MULTI-AGENT REINFORCEMENT LEARNING RESULTS

We chose the game *Mortal Kombat*, a two character side viewed fighting game (a screenshot of the game can be seen in Figure (1), as a testbed for the above, as it exhibits favorable properties: both players share the same screen, the agent's optimal policy is heavily dependent on the rival's behavior, unlike racing games for example. In order to evaluate two agents fairly, both were trained using the same characters maintaining the identity of rival and agent. Furthermore, to remove the impact of the starting positions of both agents on their performances, the starting positions were initialized randomly.

In the first experiment we evaluated all combinations of DQN against D-DQN and Dueling D-DQN. Each agent was trained against the in-game AI until convergence. Then 50 matches were performed between the two agents. DQN lost 28 out of 50 games against Dueling D-DQN and 33 against D-DQN. D-DQN lost 26 time to Dueling D-DQN. This win balance isn't far from the random case, since the algorithms converged into a policy in which movement towards the opponent is not

required rather than generalize the game. Therefore, in many episodes, little interaction between the two agents occur, leading to a semi-random outcome.

In our second experiment, we continued the training process of a the D-DQN network by letting it compete against the Dueling D-DQN network. We evaluated the re-trained network by playing 30 episodes against the in-game AI. After training, D-DQN was able to win 28 out of 30 games, yet when faced again against the in-game AI its performance deteriorated drastically (from an average of 17000 to an average of -22000). This demonstrated a form of catastrophic forgetting (Goodfellow et al., 2013) even though the agents played the same game.

In our third experiment, we trained a Dueling D-DQN agent against three different rivals: the in-game AI, a trained DQN agent and a trained Dueling-DQN agent, in an alternating manner, such that in each episode a different rival was playing as the opponent with the intention of preventing the agent from learning a policy suitable for just one opponent. The new agent was able to achieve a score of 162,966 (compared to the "normal" dueling D-DQN which achieved 169,633). As a new and objective measure of generalization, we've configured the in-game AI difficulty to be "very hard" (as opposed to the default "medium" difficulty). In this metric the alternating version achieved 83,400 compared to -33,266 of the dueling D-DQN which was trained in default setting. Thus, proving that the agent learned to generalize to other policies which weren't observed while training.

## 4.4 FUTURE CHALLENGES

As demonstrated, RLE presents numerous challenges that have yet to be answered. In addition to being able to learn all available games, the task of learning games in which reward delay is extreme, such as F-Zero without reward shaping, remains an unsolved challenge. Additionally, some games, such as Super Mario, feature several stages that differ in background and the levels structure. The task of generalizing platform games, as in learning on one stage and being tested on the other, is another unexplored challenge. Likewise surpassing human performance remains a challenge since current state-of-the-art algorithms still struggling with the many SNES games.

## 5 CONCLUSION

We introduced a rich environment for evaluating and developing reinforcement learning algorithms which presents significant challenges to current state-of-the-art algorithms. In comparison to other environments RLE provides a large amount of games with access to both the screen and the in-game state. The modular implementation we chose allows extensions of the environment with new consoles and games, thus ensuring the relevance of the environment to RL algorithms for years to come (see Table (2)). We've encountered several games in which the learning process is highly dependent on the reward definition. This issue can be addressed and explored in RLE as reward definition can be done easily. The challenges presented in the RLE consist of: 3D interpretation, delayed reward, noisy background, stochastic AI behavior and more. Although some algorithms were able to play successfully on part of the games, to fully overcome these challenges, an agent must incorporate both technique and strategy. Therefore, we believe, that the RLE is a great platform for future RL research.

## 6 ACKNOWLEDGMENTS

The authors are grateful to the Signal and Image Processing Lab (SIPL) staff for their support, Alfred Agrell and the LibRetro community for their support and Marc G. Bellemare for his valuable inputs.

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

# Appendices

Experimental Results

Table 3: Average results of *DQN*, *D-DQN*, *Dueling D-DQN* and a Human player

|               | DQN   | D-DQN | Dueling D-DQN | Human  |
|---------------|-------|-------|---------------|--------|
| F-Zero        | 3116  | 3636  | 5161          | 6298   |
| Gradius III   | 7583  | 12343 | 16929         | 24440  |
| Mortal Kombat | 83733 | 56200 | 169300        | 132441 |
| Super Mario   | 11765 | 16946 | 20030         | 36386  |
| Wolfenstein   | 100   | 83    | 40            | 2952   |

