# Peer review of "Playing SNES in the Retro Learning Environment"

_ICLR 2017 — rejected_

[Official Review · AnonReviewer3 · rating 7 · confidence 4 · 16 Dec 2016]
**No Title**

This paper presents a valuable new collection of video game benchmarks, in an extendable framework, and establishes initial baselines on a few of them.

Reward structures: for how many of the possible games have you implemented the means to extract scores and incremental reward structures? From the github repo it looks like about 10 -- do you plan to add more, and when?

“rivalry” training: this is one of the weaker components of the paper, and it should probably be emphasised less. On this topic, there is a vast body of (uncited) multi-agent literature, it is a well-studied problem setup (more so than RL itself). To avoid controversy, I would recommend not claiming any novel contribution on the topic (I don’t think that you really invented “a new method to train an agent by enabling it to train against several opponents” nor “a new benchmarking technique for agents evaluation, by enabling them to compete against each other, rather than playing against the in-game AI”). Instead, just explain that you have established single-agent and multi-agent baselines for your new benchmark suite.

Your definition of Q-function (“predicts the score at the end of the game given the current state and selected action”) is incorrect. It should read something like: it estimates the cumulative discounted reward that can be obtained from state s, starting with action a (and then following a certain policy).

Minor:
* Eq (1): the Q-net inside the max() is the target network, with different parameters theta’
* the Du et al. reference is missing the year
* some of the other references should point at the corresponding published papers instead of the arxiv versions

[Official Review · AnonReviewer2 · rating 4 · confidence 4 · 17 Dec 2016]
**Ok but limited contributions**

This paper introduces a new reinforcement learning environment called « The Retro Learning Environment”, that interfaces with the open-source LibRetro API to offer access to various emulators and associated games (i.e. similar to the Atari 2600 Arcade Learning Environment, but more generic). The first supported platform is the SNES, with 5 games (more consoles and games may be added later). Authors argue that SNES games pose more challenges than Atari’s (due to more complex graphics, AI and game mechanics). Several DQN variants are evaluated in experiments, and it is also proposed to compare learning algorihms by letting them compete against each other in multiplayer games.

I like the idea of going toward more complex games than those found on Atari 2600, and having an environment where new consoles and games can easily be added sounds promising. With OpenAI Universe and DeepMind Lab that just came out, though, I am not sure we really need another one right now. Especially since using ROMs of emulated games we do not own is technically illegal: it looks like this did not cause too much trouble for Atari but it might start raising eyebrows if the community moves to more advanced and recent games, especially some Nintendo still makes money from.

Besides the introduction of the environment, it is good to have DQN benchmarks on five games, but this does not add a lot of value. The authors also mention as contribution "A new benchmarking technique, allowing algorithms to compete against each other, rather than playing against the in-game AI", but this seems a bit exaggerated to me: the idea of pitting AIs against each other has been at the core of many AI competitions for decades, so it is hardly something new. The finding that reinforcement learning algorithms tend to specialize to their opponent is also not particular surprising.

Overall I believe this is an ok paper but I do not feel it brings enough to the table for a major conference. This does not mean, however, that this new environment won't find a spot in the (now somewhat crowded) space of game-playing frameworks.

Other small comments:
- There are lots of typos (way too many to mention them all)
- It is said that Infinite Mario "still serves as a benchmark platform", however as far as I know it had to be shutdown due to Nintendo not being too happy about it
- "RLE requires an emulator and a computer version of the console game (ROM file) upon initialization rather than a ROM file only. The emulators are provided with RLE" => how is that different from ALE that requires the emulator Stella which is also provided with ALE?
- Why is there no DQN / DDDQN result on Super Mario?
- It is not clear if Figure 2 displays the F-Zero results using reward shaping or not
- The Du et al reference seems incomplete

[Official Review · AnonReviewer1 · rating 5 · confidence 4 · 19 Dec 2016 (modified: 23 Jan 2017)]
**Final review: Nice software contribution, expected more significant scientific contributions**

The paper presents a new environment, called Retro Learning Environment (RLE), for reinforcement learning. The authors focus on Super Nintendo but claim that the interface supports many others (including ALE). Benchmark results are given for standard algorithms in 5 new Super Nintendo games, and some results using a new "rivalry metric".

These environments (or, more generally, standardized evaluation methods like public data sets, competitions, etc.) have a long history of improving the quality of AI and machine learning research. One example in the past few years was the Atari Learning Environment (ALE) which has now turned into a standard benchmark for comparison of algorithms and results. In this sense, the RLE could be a worthy contribution to the field by encouraging new challenging domains for research.

That said, the main focus of this paper is presenting this new framework and showcasing the importance of new challenging domains. The results of experiments themselves are for existing algorithms. There are some new results that show reward shaping and policy shaping (having a bias toward going right in Super Mario) help during learning. And, yes, domain knowledge helps, but this is obvious. The rivalry training is an interesting idea, when training against a different opponent, the learner overfits to that opponent and forgets to play against the in-game AI; but then oddly, it gets evaluated on how well it does against the in-game AI! 

Also the part of the paper that describes the scientific results (especially the rivalry training) is less polished, so this is disappointing. In the end, I'm not very excited about this paper.

I was hoping for a more significant scientific contribution to accompany in this new environment. It's not clear if this is necessary for publication, but also it's not clear that ICLR is the right venue for this work due to the contribution being mainly about the new code (for example, mloss.org could be a better 'venue', JMLR has an associated journal track for accompanying papers:

[Final Decision · Program Chairs · 06 Feb 2017]
**ICLR committee final decision**

The authors present a new set of environments, similar to ALE but based on Super Nintendo rather than Atari. This is a great asset and could be important for RL research, but it doesn't merit ICLR publication because of the lack of novel research ideas. Hopefully the authors will consider another venue to publish this paper, such as perhaps a journal or workshop.